# Analysis of Photosynthetic Characteristics and Screening High Light-Efficiency Germplasm in Sugarcane

**DOI:** 10.3390/plants13050587

**Published:** 2024-02-22

**Authors:** Yibin Wei, Yuzhi Xu, Abdullah Khan, Chunxiu Jiang, Huojian Li, Yuling Wu, Chi Zhang, Maoyao Wang, Jun Chen, Lifang Zeng, Muqing Zhang

**Affiliations:** 1College of Agriculture, Guangxi University, Nanning 530004, China; 2117301048@st.gxu.edu.cn (Y.W.);; 2Guangxi Key Laboratory of Sugarcane Biology & State Key Laboratory for Conservation and Utilization of Subtropical Agro-Bioresources, Guangxi University, Nanning 530004, China

**Keywords:** sugarcane, photosynthetic characteristics, genotype, germplasm evaluation

## Abstract

Sugarcane is a globally significant crop for sugar and energy production, and developing high light-efficiency sugarcane varieties is crucial for enhancing yield and quality. However, limited research is available on the screening of sugarcane germplasm with high photosynthetic efficiency, especially with different leaf positions. The present study, conducted in Guangxi, China, aimed to analyze the photosynthetic characteristics of 258 sugarcane varieties at different leaf positions over three consecutive years in field experiments. The results showed significant differences in photosynthetic characteristics among genotypes, years, and leaf positions. Heritability estimates for various photosynthetic parameters ranged from 0.76 to 0.88. Principal component analysis revealed that the first three principal components accounted for over 99% of the cumulative variance. The first component represented photosynthetic efficiency and light utilization, the second focused on electron transfer and reaction center status, and the third was associated with chlorophyll content. Cluster and discriminant analysis classified sugarcane genotypes into three categories: high photosynthetic efficiency (HPE) with 86 genotypes, medium photosynthetic efficiency (MPE) with 60 genotypes, and low photosynthetic efficiency (LPE) with 112 genotypes. Multi-year trials confirmed that HPE sugarcane genotypes had higher single-stem weight and sucrose content. This study provides valuable insights into the photosynthetic physiological characteristics of different sugarcane varieties, which can contribute to further research regarding high yields and sugar breeding.

## 1. Introduction

Sugarcane (*Saccharum officinarum* L.) is a globally cultivated economic crop responsible for over 80% of global sugar and 40% of alcohol production [1,2]. Enhancing sugarcane yield and sugar content has been a longstanding challenge in the sugarcane industry [3]. Global climate change has the potential to disrupt temperature and precipitation patterns, resulting in adverse effects on the growth of sugarcane [4,5]. In response, improving photosynthetic efficiency has become a crucial strategy to help plants cope with climate change, ultimately enhancing both the yield and quality of sugarcane [5]. Moreover, the scarcity of resources, including water and fertilizers, poses a significant challenge to the sugarcane industry. By optimizing photosynthetic efficiency, we can effectively mitigate the adverse impacts of climate change on the sugarcane industry. This initiative helps reduce the demand for limited resources like water and fertilizers, enhancing their utilization efficiency [6]. Historically, agriculture has predominantly depended on pesticides to boost global agricultural production [7,8]. However, this prevalent approach has negative consequences, including soil pollution, biodiversity loss, and human health threats [9]. The agricultural industry is increasingly prioritizing sustainability. Elevating the photosynthetic efficiency of sugarcane not only alleviates pressure on water resources but also contributes to more sustainable cultivation practices [10]. This is particularly crucial for sustaining the production of high-quality sugarcane on limited land.

Elevating photosynthetic efficiency signifies a more effective conversion of light energy into chemically usable energy for plants. We can maximize the productivity of sugarcane plants by optimizing key steps in photosynthesis, such as improving the utilization efficiency of photosynthetic pigments and enhancing the efficiency of the electron transfer chain [11]. This optimization increases yields and is often accompanied by enhanced plant resilience to environmental stresses. This is particularly crucial for sugarcane, as its growth might be influenced by various factors such as weather, soil conditions, and diseases. Improving photosynthetic efficiency enables sugarcane plants to better adapt to changing environments, including coping with climate variations, soil fluctuations, and disease pressures [12,13]. This effort contributes to strengthening the resilience of sugarcane, enhancing yields, and improving quality [14]. This might involve breeding varieties with higher photosynthetic efficiency and adjusting growth environments to optimize the photosynthetic process. Therefore, enhancing the photosynthetic efficiency of sugarcane is not only an effective means of increasing yields but also a crucial step in making the crop more adaptive and robust [15,16]. It is essential for ensuring the sustainable development of the sugarcane industry under constantly changing environmental conditions. Approximately 90% of organic matter accumulation stems from photosynthesis, making photosynthetic efficiency and intensity critical for crop production [17]. As a C4 plant, sugarcane photosynthetic potential has yet to be fully harnessed [18], offering an opportunity to increase yield [19]. Consequently, enhancing sugarcane photosynthesis has become a potential strategy for productivity improvement [14,20].

Chlorophyll fluorescence is the re-emission of light from chlorophyll after it has absorbed energy and returned from an excited state to the ground state; this optical phenomenon has been established as a crucial indicator for measuring photosynthesis, particularly the efficiency of photosystem II (PSII) [21]. Photosynthesis involves complex primary processes such as light capture, transfer, and conversion [22]. Photosystem II (PSII) is sensitive to light damage and impacts photosynthesis, making its primary reactions essential for the entire process [23]. Thus, chlorophyll fluorescence has become a precious tool for monitoring PSII functionality and assessing overall photosynthesis, such as light utilization efficiency during various crop growth stages [24]. Evaluating photosynthetic performance is crucial for improving sugarcane yield and quality [25], and chlorophyll fluorescence kinetics techniques can rapidly and non-destructively reveal photosynthetic characteristics within leaves [21]. When plants are exposed to adverse environmental conditions for an extended period, chlorophyll content decreases and photosynthesis rates drop, leading to changes in fluorescent parameters [26]. Chlorophyll fluorescence parameters are crucial for assessing plant photosynthetic efficiency [27]. Measuring parameters such as maximum photochemical efficiency (Fv/Fm) allows us to understand the potential of plant photosynthesis, aiding in the study of the impact of environmental stress on plant physiology and optimizing plant growth conditions [28,29,30]. Chlorophyll fluorescence measurements simultaneously provide information about the physiological status of plants, assisting in diagnosing the physiological state of plants under stress conditions [31]. In ecological research, a comprehensive analysis of chlorophyll fluorescence parameters contributes to understanding the health of plant communities and ecosystems [32]. One can infer plant adaptability, growth status, and dynamic changes in ecosystems by comparing chlorophyll fluorescence in different species or growth environments. In agriculture, chlorophyll fluorescence parameters are analyzed to monitor crop growth and predict yields. This aids in optimizing field management, adjusting irrigation and fertilization strategies, and ultimately enhancing crop yield and quality [33]. Simultaneously, chlorophyll fluorescence is an indicator in environmental monitoring, assessing plant responses to environmental changes. Monitoring chlorophyll fluorescence helps to understand plant adaptability to pollution, climate change, and soil conditions [28,34,35,36]. The rapid detection and widespread application of fluorescence parameters like Fv/Fm (PSII maximum photochemical efficiency) and Y(NO) (nonphotochemical quenching quantum yield) provide technical support for efficient breeding, which not only aids in selecting high-efficiency sugarcane varieties but also deepens our understanding of their stress adaptability, photosynthetic efficiency, and genetic basis [37].

Screening germplasm for enhanced photosynthetic performance allows the identification and selection of varieties with superior carbon assimilation capabilities, essential for optimizing biomass production and increasing crop yields. Therefore, the present study was conducted to comprehensively analyze chlorophyll fluorescence parameters and relative chlorophyll content at different leaf positions of 258 sugarcane germplasms over three years and to elucidate the photosynthetic characteristics of sugarcane genotypes and provide a theoretical basis for high-efficiency sugarcane breeding.

## 2. Results

### 2.1. Combined Variance Analysis for Sugarcane Photosynthetic Characteristics

From 2021 to 2023, various photosynthetic parameters were investigated on different sugarcane genotypes. Variance analysis revealed significant differences for genotype (G) (*p* < 0.001), year (Y) (*p* < 0.001), leaf position (L) (*p* < 0.001), and their interactions, indicating highly significant differences in photosynthetic efficiency (Table 1). There was a significant interaction between G × Y and G × L (*p* < 0.001). However, the differences between replicates were not significant in Fv, Fv/Fo, Fv/Fm, Y(NO), and SPAD, indicating good repeatability of the measurements. Parameters exhibited genotype differences in Fv/Fm, Fv/Fo, Y(NO), and SPAD, suggesting that sugarcane photosynthetic performance varies with genotype (Table 1). The phenotypic data from all three years were normally distributed (Figure 1). Chlorophyll content in newly planted sugarcane (NP-2021) was significantly higher than that in the first ratooning (R1-2022) and the second ratooning (R2-2023) sugarcane. Chlorophyll fluorescence parameters Fo, Fm, and Fv were higher in the first ratooning sugarcane than in newly planted and the second ratooning sugarcane (Figure 1A–G), indicating that sugarcane photosynthesis is influenced by both genotype and environment.

The photosynthetic parameters and chlorophyll content significantly increased as the leaf position increased from +1 to +3 in the canopy, including Fm, Fv, Fo, Fv/Fm, and Fv/Fo. At the same time, Y(NO) exhibited the opposite trend, with increasing leaf position (Figure 1H–N). These results suggested that leaf functionality gradually improves and photosynthetic capacity increases from +1 to +3 leaf. The decrease in Y(NO) implied that leaves at higher positions could more efficiently convert absorbed light energy into photosynthetic products, thus reducing energy wastage and indicating that higher-positioned leaves were more mature and efficient in light energy utilization.

### 2.2. Principal Component Analysis for Sugarcane Photosynthetic Efficiency

Following standardization, principal component analysis was conducted by analyzing sugarcane photosynthetic parameters and obtaining the means of the data over three consecutive years (2021–2023). The results indicated that the first three principal components effectively represented most of the phenotypic variation in the original data, with a cumulative contribution rate exceeding 99% (Table 2).

The first principal component included parameters of Fo, Fv/Fm, Fv/Fo, and Y(NO), suggesting that the overall photosynthetic performance helps us understand the photosynthetic capacity of the leaves. The second principal component primarily comprised Fm and Fv, which typically reflected the electron transfer in leaf photosynthesis and the state of reaction centers. The third principal component included SPAD, commonly used to indicate chlorophyll content.

### 2.3. Cluster Analysis and Discriminant Analysis for Sugarcane Photosynthetic Efficiency

Cluster and discriminant analysis results revealed that only five genotypes were misclassified, with an average accuracy of 98.1% (Appendix A). Ultimately, the 258 sugarcane genotypes were categorized into three groups: high photosynthetic efficiency (HPE), moderate photosynthetic efficiency (MPE), and low photosynthetic efficiency (LPE). Each category represented 33.3%, 23.3%, and 43.4% of the total genotypes, respectively (Figure 2). Further multiple comparison analysis showed significant differences in photosynthetic efficiency indicators between these categories (Table 3), which helped better understand the differences in photosynthetic efficiency among different categories of sugarcane genotypes.

Genotypes with high photosynthetic efficiency exhibited excellent performance, mainly due to their advantages in key fluorescence parameters and chlorophyll content. High levels of Fv/Fm and Fv/Fo indicated their efficient photosystem II (PSII) photochemical efficiency, suggesting that they could convert light energy into biochemical energy more effectively. Additionally, the increase in SPAD values indicated a high chlorophyll content, which contributed to enhanced photosynthetic efficiency. Low Y(NO) levels suggested their ability to minimize light energy loss more effectively, possibly related to their more robust nonphotochemical quenching mechanisms.

LPE genotypes performed poorly in multiple parameters, mainly because their photosynthetic capacity was limited. Higher Fo values indicated more initial fluorescence in PSII, possibly due to lower electron density in PSII reaction centers. Furthermore, lower Fm, Fv, Fv/Fm, and Fv/Fo indicated issues with electron transfer and the state of PSII, resulting in lower light energy conversion efficiency. High Y(NO) levels suggested relatively high light energy losses, possibly because PSII could not fully dissipate the absorbed light energy.

MPE genotypes exhibited average performance in some parameters, indicating moderate photosynthetic capability. High Fo and Fm values suggested moderate initial and maximum fluorescence, while a high Fv value indicated relatively balanced electron transfer in PSII. Simultaneously, moderate levels of Fv/Fm, Fv/Fo, and Y(NO) suggested that their light energy conversion efficiency and quenching mechanisms performed moderately compared to genotypes with high photosynthetic efficiency. The lower SPAD values indicated lower chlorophyll content, which could be one of the reasons affecting their photosynthetic performance.

### 2.4. Quality Efficiency of Sugarcane with Different Photosynthetic Traits

Data were collected over two years of field experiments on the single-stalk weight and sucrose content of each genotype. The results indicated that HPE sugarcane germplasm exhibited higher single-stalk weight and sucrose content (Figure 3), suggesting that HPE plants could utilize light energy more effectively for photosynthesis, resulting in the production of more nutrients and energy, leading to higher single-stalk weight and sucrose content. Conversely, LPE germplasm exhibited the opposite trend. Genotypes with moderate photosynthetic efficiency had single-stalk weight and sucrose content values that fell between the extremes, indicating that high photosynthetic efficiency might be a critical factor in increasing sugarcane yield and quality. This finding has important implications for sugarcane cultivation and breeding.

## 3. Discussion

The results presented here showed that both genotype and environment significantly affected sugarcane photosynthesis over three years. Through a combined variance analysis, the differences in sugarcane photosynthetic traits significantly differed over multiple years and leaf positions on each sugarcane genotype. Our results indicated significant chlorophyll content and fluorescence variations across years, leaf positions, and interactions. Previous research has shown that environmental factors, leaf traits, and internal structures significantly affect plant photosynthetic activity, thereby emphasizing the importance of season, climate, and canopy structure on photosynthetic efficiency [38,39,40,41]. The weight for genetic components in sugarcane photosynthetic efficiency varied between 15.055% and 28.498%, indicating that genetic factors primarily control photosynthetic traits in sugarcane [42]. Additionally, adverse environmental conditions could reduce the chlorophyll content of sugarcane leaves, depending on the genetic differences among sugarcane genotypes [43,44]. Therefore, assessing sugarcane photosynthetic traits across multiple environments is essential for improving breeding accuracy.

Chlorophyll content and fluorescence parameters have been widely employed in studying plant photosynthetic traits [45]. The present study examined sugarcane photosynthetic efficiency over three consecutive years. PCA analysis showed that the first three principal components contribute more than 99% of the variance (Table 2). In previous studies, PCA has been widely used in exploring the correlations between different variables [46,47,48]. PCA strategies for complex phenotypes have successfully established phenotype variables for GWAS analysis [49] and to explain the natural variation in photosynthetic traits among different African rice populations [50]. SPAD, an independent assessment parameter, has also been widely applied in various fields, including varietal selection, trait evaluation, and plant performance under biotic and abiotic stress conditions [51,52,53]. The results in Table 2 highlight the significant roles of Fo, Fv/Fm, Fv/Fo, and Y(NO) in sugarcane photosynthetic parameter variations, as they play critical roles in the first principal component, explaining 48.9% of the total variance. These parameters reflect the efficiency of chlorophyll fluorescence in converting light energy in chlorophyll molecules and the performance in electron transfer processes. Fo, the initial fluorescence, shows the electron density in photosystem II (PSII). Fv/Fm and Fv/Fo, on the other hand, reflect the maximum photochemical efficiency of PSII, representing the efficiency of the primary conversion of light energy in PSII and providing information on plants’ potential maximum photosynthetic capacity. Y(NO) reflects the extent of light damage, with a higher Y(NO) indicating incomplete absorption and conversion of light energy [54,55].

Furthermore, the study demonstrated the significant impact of Fm and Fv on the variation of sugarcane photosynthetic efficiency, as they are the major loading factors in the second principal component, explaining 37.46% of the variance. Fm and Fv typically reflect the status of electron transfer and reaction centers in photosynthesis. This can help us better understand the electron transfer processes in photosynthesis and the possible electron transfer efficiency. Therefore, the importance of these parameters highlights the critical roles of Fo, Fv/Fm, Fv/Fo, Y(NO), Fm, and Fv in sugarcane physiology, which not only aids in understanding the photosynthetic adaptability and performance of sugarcane plants but also provides a theoretical basis for optimizing the study of photosynthesis [56,57].

Photosynthetic efficiency is a crucial physiological process in plants carried out under illumination and is controlled by multiple genes [58]. Breeders have identified the genetic basis of different photosynthetic efficiencies in sugarcane, including photosynthetic rates, chlorophyll content [59], and photosynthetic electron transfer rates [60]. This study found that the variation in sugarcane photosynthetic efficiency is primarily attributed to genetic factors, as shown in Table 1. The clustering results revealed that HPE genotypes, such as ROC22, exhibit significant differences in their offspring’s photosynthetic traits when used as a parent in various combinations. For instance, the offspring of combinations like CT89-103 × ROC22 and YT94-128 × ROC22 belong to different photosynthetic efficiency groups (Appendix A), suggesting that the photosynthetic traits of offspring could be influenced when ROC22 serves as a parent and is crossed with other varieties. This phenomenon might result from parental combinations’ effect on the offspring’s photosynthetic efficiency [61,62], leading to different photosynthetic traits [63]. The offspring from the parental combination ROC22 (HPE) and GT92-66 (MPE) exhibit significant superiority in photosynthetic traits. For example, the offspring of GT42 (ROC22 × GT92-66) and 14-2802 (GT92-66 × ROC22), both belonging to high photosynthetic efficiency groups, demonstrate better photosynthetic performance, suggesting that these offspring exhibit higher performance in photosynthetic traits, likely due to the genetic advantages brought by the combination of these two parents, ROC22 (HPE) and GT92-66 (MPE). This combination potentially introduced a range of favorable genetic factors for photosynthesis, including higher photosynthetic efficiency, increased chlorophyll content, or other physiological traits related to photosynthesis. The interaction and complementarity of these genetic factors may significantly improve the photosynthetic traits of the offspring, giving them more potential in terms of growth and yield [63,64]. Thus, the parental combination ROC22 (HPE) and GT92-66 (MPE) provides promising germplasm materials for breeding projects with the potential to cultivate varieties with higher photosynthetic efficiency and superior yield performance, offering excellent prospects for improving and optimizing sugarcane. This finding further underscores the critical role of genetic combinations in breeding and how selecting the right parental combinations can achieve desired breeding goals.

Most genotypes with high and moderate photosynthetic efficiency were developed through direct or indirect crosses with high-quality American parental lines. Conversely, LPE genotypes were predominantly obtained using introduced varieties as parents (Appendix A). Introduced materials typically possess diverse geographical, ecological, and genetic backgrounds, which might confer favorable adaptive traits to local environments [65]. Hybridizing these introduced materials with local varieties increases the chances of genetic diversity in sugarcane, enhancing the population’s adaptability to different environmental conditions [66] and contributing to broader adaptability and improved growth performance in various regions [67]. Combining introduced germplasms with local varieties promotes the incorporation of beneficial genetic factors in the offspring, leading to improvements in specific traits [68], such as enhanced resistance to biotic and abiotic stressors in sugarcane [69,70]. Genetic improvement could achieve superior performance in multiple aspects of new varieties. Introducing new genotypes enriches genetic diversity, providing breeders with a broader selection and allowing them to search for genotypes in the germplasm collections that suit their specific needs regarding traits and efficiency [71,72]. In summary, hybridizing introduced materials with local varieties combines genetic resources and ecological traits to enhance and optimize sugarcane varieties. This approach increases productivity, adaptability, and resistance in newly released varieties, promoting sustainable agricultural development.

## 4. Materials and Methods

### 4.1. Experimental Location and Design

This study was carried out for three consecutive years in February 2021 in the Agricultural Field Station of Guangxi University, Fusui, China (107°78′ E, 22°51′ N), including new planting (NP, 2021), the first ratooning (R1, 2022), and the second ratooning (R2, 2023). The local conditions featured an altitude of 69.5 m, an average annual sunshine duration of 1693 h, total radiation of 108.4 Kcal/cm, an average frost-free period of 346 days, and a rainfall range of 1050–1300 mm. A total of 258 sugarcane genotypes introduced and bred in China were selected with the detailed information provided in the Appendix A. Each genotype was propagated from healthy and vigorous sugarcane buds. A completely randomized block design with three replicates was employed, with 16 sugarcane buds planted per meter of a single row. Each row measured 5 m in length with 1.4 m row spacing. Two widely cultivated sugarcane cultivars of ROC22 and GT42 in China were included as control varieties. The experimental field was performed in red soil, and its physicochemical properties were as follows: pH 5.70; organic matter 18.50 g/kg; total nitrogen content 0.70 g/kg; available phosphorus content 218 mg/kg; available potassium content 81.40 mg/kg; and alkaline hydrolysis nitrogen 95.10 mg/kg. New planting and ratooning management for sugarcane were consistent with local field management.

### 4.2. Field Data Collection

Six months after sugarcane planting or harvesting, when the sugarcane was in the elongation phase, data were collected for seven consecutive days during clear weather conditions. The procedure involved measuring fluorescence parameters using a PAM-2500 portable modulated chlorophyll fluorometer (Walz, Germany) on the central sections of leaves from +1 leaf (the first fully unfolded leaf of the cane) to +3 leaf on healthy sugarcane plants (Appendix A). The chlorophyll fluorescent parameters mainly included:

Fo (initial fluorescence) represents the fluorescence intensity when the photosystem II reaction center is fully open, reflecting the electron density excited by the antenna pigment of photosystem II related to chlorophyll concentration. For high-efficiency germplasm, Fo values are typically lower, indicating more effective capture of light energy by chlorophyll antenna pigments.

Fm (maximum fluorescence) represents the fluorescence intensity when the photosystem II reaction center is fully closed, indicating the electron transfer in photosystem II. The germplasm with high-photosynthetic efficiency (HPE) usually has higher Fm values, indicating more efficient electron transfer in photosystem II.

Fv (variable fluorescence), the maximum variable fluorescence intensity in the dark, is calculated by subtracting Fo from Fm, which reflects the reduction state of the primary electron acceptors of PSII. Fv values are typically higher for HPE germplasm, indicating that the primary electron acceptors are more easily reduced, enhancing the utilization efficiency of light energy.

Fv/Fm (variable fluorescence/maximum fluorescence) reflects the maximum quantum yield of PSII, representing the plant’s potential maximum photosynthetic capacity. The HPE germplasm typically has Fv/Fm values close to or greater than 0.82, indicating a higher photosynthetic capacity.

Fv/Fo (variable fluorescence/initial fluorescence) reflects the electron transfer in PSII and the potential activity of PSII. A higher Fv/Fo indicates more efficient electron transfer and higher potential activity of PSII. The HPE germplasm usually has higher Fv/Fo values, showing higher electron transfer efficiency in PSII.

Y(NO) (non-regulatory quenching quantum yield), an indicator of photodamage, indicates that the photochemical energy conversion and protective regulatory mechanisms are insufficient to dissipate absorbed light energy fully. The HPE germplasm typically has lower Y(NO) values, indicating more effective light energy utilization and reduced energy wastage [73].

Additionally, during the daytime at the same period, chlorophyll content in the central sections of leaves was measured from +1 leaf to +3 leaf of healthy sugarcane plants from different genotypes using a SPAD-502 Plus instrument (Konica Minolta, Tokyo, Japan), typically ranging between 30 and 60 SPAD units.

### 4.3. Sugarcane Single-Stem Weight and Sugar Content Measurement

In December 2021 and December 2022, during the sugarcane maturity period, ten representative sugarcane samples were collected for each genotype. The samples were cleaned by removing leaf sheaths and impurities, and their weight was measured using an electronic scale. The single-stem weight for each genotype was calculated. Subsequently, the fresh sugarcane stems collected were processed using the DM540-CPS system (Sugarcane Crushing System, IRBI, Brazil). Simultaneously, near-infrared spectral data were collected using the MA TRIX-F system (Bruker Optik GmbH, Germany) [74].

### 4.4. Statistical Analysis

Phenotypic data for sugarcane were recorded and processed using Excel 2021. Subsequently, a mixed-effects analysis was conducted using the “lme4” package in R. Next, principal component analysis (PCA) was performed on the mean values of photosynthetic efficiency indicators for different years and all years combined, using the “FactoMineR” package in R. Furthermore, cluster and discriminant analyses were carried out using SPSS 2022 with the principal component scores, and posterior probabilities were calculated. Multiple photosynthetic efficiency indicators were compared between different categories using the LSD method. Charts and graphs were created using Origin 2022 software.

## 5. Conclusions

This study comprehensively analyzed the photosynthetic efficiency of 258 Chinese-introduced and bred sugarcane genotypes. The results emphasized the significant impact of different years and leaf positions on sugarcane photosynthetic efficiency and notable interactions between genotype and environmental factors. Principal component analysis elucidated the critical role of chlorophyll fluorescence parameters such as Fo, Fv/Fo, Fv/Fm, and Y(NO) in explaining the variance in sugarcane photosynthetic efficiency. Through cluster analysis and discriminant analysis, sugarcane genotypes were classified into three categories: high photosynthetic efficiency (HPE) with 86 genotypes, moderate photosynthetic efficiency (MPE) with 60 genotypes, and low photosynthetic efficiency (LPE) with 112 genotypes. The study also revealed that high photosynthetic efficiency in sugarcane germplasm is often associated with higher single-stalk weight and sucrose content, providing a potential strategy for improving sugarcane yield and quality.

## Figures and Tables

**Figure 1 plants-13-00587-f001:**
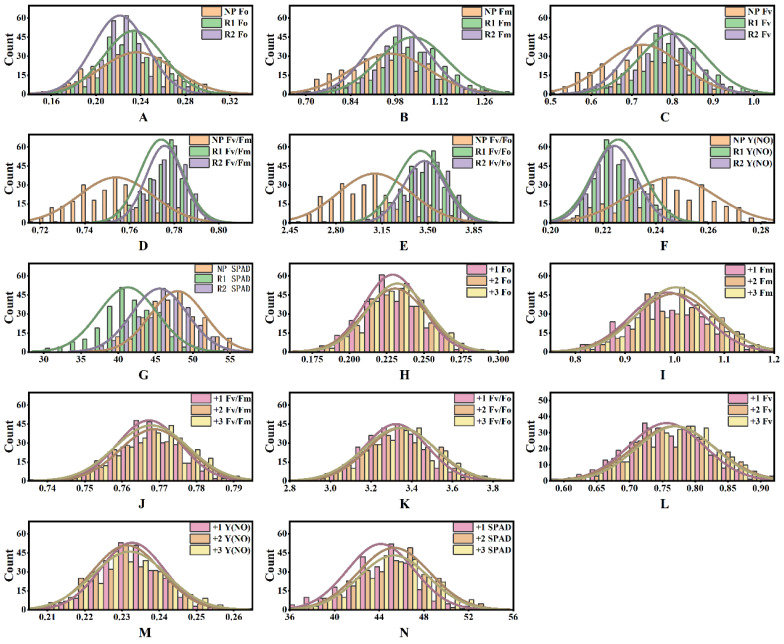
Descriptive statistics of photosynthetic efficiency of sugarcane. (**A**–**G**) The difference and distribution of photosynthetic efficiency of sugarcane in different years; (**H**–**N**) the difference and distribution of photosynthetic efficiency of sugarcane at different leaf positions. The number indicates the leaf position.

**Figure 2 plants-13-00587-f002:**
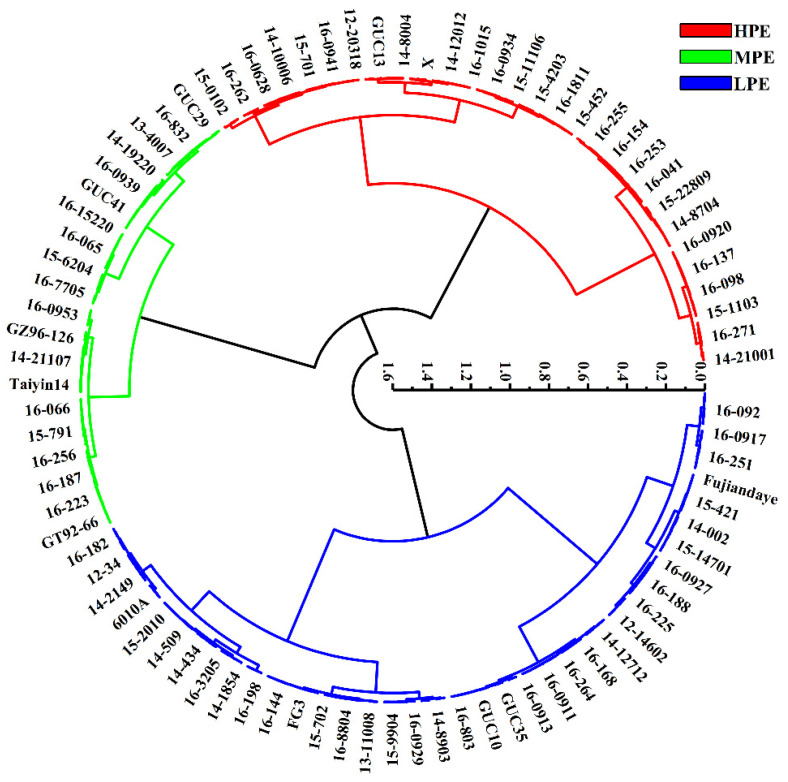
Cluster for photosynthetic characteristic parameters in 258 sugarcane genotypes. Red represents 86 germplasms with high photosynthetic efficiency (HPE), green represents 60 germplasms with medium photosynthetic efficiency (MPE), and blue represents 112 germplasms with low photosynthetic efficiency (LPE).

**Figure 3 plants-13-00587-f003:**
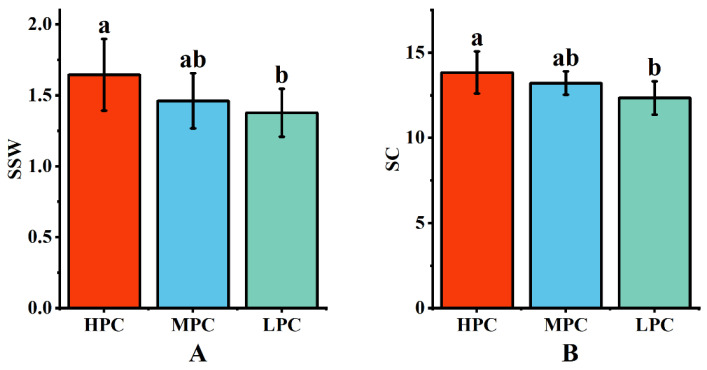
Relationship between yield and quality of sugarcane with different photosynthetic efficiencies. (**A**) Single-stem weight of sugarcane taxa with different photosynthetic efficiencies; (**B**) sucrose content of sugarcane taxa with different photosynthetic efficiencies. Different letters indicate significant differences among the genotypes (LSD test, *p* < 0.05).

**Table 1 plants-13-00587-t001:** Combined variance analysis for chlorophyll fluorescence parameters and relative chlorophyll content.

Source of Variation	df	Fm		Fo		Fv		Fv/Fm		Fv/Fo		Y(NO)		SPAD	
		Mean Square	SS (%)	Mean Square	SS (%)	Mean Square	SS (%)	Mean Square	SS (%)	Mean Square	SS (%)	Mean Square	SS (%)	Mean Square	SS (%)
Genotype (G)	257	0.1271 ***	20.1	0.00905 ***	23.0	0.076 ***	19.0	0.0019 ***	18.6	0.62 ***	19.5	0.0019 ***	18.6	236 ***	28.5
Year (Y)	2	3.0716 ***	3.8	0.15122 ***	3.0	3.215 ***	6.3	0.3431 ***	26.2	102.73 ***	25.2	0.3431 ***	26.2	26540 ***	25.0
Leaf Position (L)	2	0.1803 ***	0.2	0.00569 ***	0.1	0.124 ***	0.23	0.0011 ***	0.1	0.44 ***	0.1	0.0011 ***	0.1	1066 ***	1.1
Rep (R)	2	0.0695 *	0.1	0.00553 *	0.1	0.036 *	0.1	0.0001	0.0	0.07	0.0	0.0001	0.0	11	0.0
G × Y	514	0.0974 ***	30.8	0.0064 ***	32.5	0.058 ***	29.0	0.001 ***	19.6	0.31 ***	19.3	0.001 ***	19.6	77 ***	18.5
G × L	514	0.0163 ***	5.6	0.00095 ***	4.9	0.01 ***	5.1	0.0002 ***	4.0	0.06 ***	4.1	0.0002 ***	4.0	12 ***	2.9
Y × L	4	0.3466 ***	0.9	0.00957 ***	0.4	0.241 ***	0.9	0.0016 ***	0.2	0.53 ***	0.3	0.0016 ***	0.2	1187 ***	2.2
G × Y × L	1028	0.014 ***	8.9	0.00076 ***	7.7	0.009 ***	9.0	0.0002 ***	6.6	0.05 **	6.4	0.0002 ***	6.6	12 ***	5.9
Residuals	4642	0.0106	30.2	0.00062	28.3	0.007	30.3	0.0001	24.7	0.04	25.0	0.0001	24.7	7	16.0
h^2^ (%)		76.13		77.97		76.06		81.69		82.63		81.69		88.3	

*** *p* ≤ 0.001; ** *p* ≤ 0.01; * *p* ≤ 0.05.

**Table 2 plants-13-00587-t002:** Principal component analysis for photosynthetic efficiency and factor weight of sugarcane.

Traits	PC1	PC2	PC3
Fm	0.507	**0.856**	−0.102
Fo	**0.838**	0.537	−0.074
Fv	0.366	**0.924**	−0.107
Fv/Fm	−**0.880**	0.473	−0.024
Fv/Fo	−**0.878**	0.475	−0.013
Y(NO)	**0.880**	−0.473	0.024
SPAD	0.103	0.266	**0.958**
Eigenvalue	3.42	2.62	0.95
Proportion of Variance	48.92	37.40	13.53
Cumulative Proportion	48.92	86.31	99.85
SS (%)	48.99	37.46	13.55

Bold digits in the table represent the loadings or weights of each variable on the corresponding principal component. The magnitude and sign of the bold digits indicate the strength and direction of the relationship between the variables and the corresponding principal components.

**Table 3 plants-13-00587-t003:** Differences in photosynthetic efficiency among different experimental genotypes.

Grade of Photosynthetic	High Photosynthetic Efficiency (HPE)	Moderate Photosynthetic Efficiency (MPE)	Low Photosynthetic Efficiency (LPE)
**Tested genotype**	6105, 24201, 09-175, 11-11319, 12-20318, 12-6403, 14-10006, 14-12012, 14-12506, 14-14707, 14-15239, 14-15418, 14-18504, 14-21001, 14-2244, 14-2802, 14-3508, 14-8004, 14-8704, 15-0102, 15-1103, 15-11106, 15-16850, 15-18106, 15-22809, 15-23304, 15-3303, 15-42, 15-4203, 15-451, 15-452, 15-4818, 15-701, 15-793, 15-W3, 16-041, 16-0628, 16-063, 16-0812, 16-084, 16-0916, 16-0920, 16-0924, 16-0930, 16-0934, 16-0941, 16-098, 16-1015, 16-104, 16-106, 16-11708, 16-12026, 16-12509, 16-1322, 16-1342, 16-137, 16-151, 16-154, 16-1811, 16-184, 16-192, 16-195, 16-2026, 16-226, 16-231, 16-253, 16-255, 16-262, 16-271, 16-401, 16-7010, 16-7506, 16-8716, 19-607, FG2, FN0335, FN10-0574, GT03-351, GT42, GT05-378, GUC13, GUC23, GZ74-141, ROC22, X, YG39	3203, 12-1801, 13-1105, 13-4007, 14-14325, 14-18509, 14-19220, 14-20701, 14-21107, 14-2720, 14-4315, 15-1005, 15-1106, 15-2007, 15-4513, 15-5404, 15-6204, 15-791, 15-794, 16-064, 16-065, 16-066, 16-0914, 16-0928, 16-0936, 16-0939, 16-0953, 16-0954, 16-12506, 16-1330, 16-136, 16-15220, 16-1612, 16-167, 16-186, 16-187, 16-222, 16-223, 16-224, 16-22402, 16-256, 16-453, 16-7019, 16-7705, 16-7722, 16-832, 16-8801, GT92-66, GT94-119, GUC17, GUC25, GUC29, GUC41, GUC8, GZ96-126, ROC27, Taiyin14, TB3, Xi096, ZZ9	3717, 6101, 8914, 35365, 40375, 11-2819, 14-15220, 06-0918, 10-228, 11-20318, 11-601, 12-106, 12-12803, 12-14602, 12-17204, 12-34, 13-11008, 13-11919, 13-14812, 13-18402, 13-21501, 14-002, 14-12712, 14-1854, 14-2149, 14-3902, 14-434, 14-509, 14-5603, 14-8009, 14-8705, 14-8903, 15-14701, 15-1743, 15-2010, 15-421, 15-453, 15-5306, 15-6008, 15-6201, 15-6402, 15-702, 15-9904, 16-043, 16-087, 16-088, 16-091, 16-0911, 16-0913, 16-0917, 16-092, 16-0926, 16-0927, 16-0929, 16-093, 16-0931, 16-0942, 16-096, 16-10002, 16-11203, 16-11905, 16-12512, 16-1329, 16-1331, 16-1335, 16-142, 16-144, 16-163, 16-168, 16-1715, 16-182, 16-188, 16-198, 16-22419, 16-225, 16-232, 16-251, 16-264, 16-3205, 16-3417, 16-5402, 16-615, 16-7719, 16-803, 16-831, 16-8701, 16-8804, 20-718, 6010A, CP01-1372, FG3, FN04-3504, Fujiandaye, Ganjiang18, GT02-390, GUC10, GUC16, GUC2, GUC21, GUC3, GUC31, GUC35, GUC7, LC05-129, ROC16, Shuidian25, TB1, TB11, YC64-389, YG24, YR03-425, YR99-596
**Fm**	1.001 ± 0.062 B	1.042 ± 0.052 A	0.963 ± 0.065 C
**Fo**	0.225 ± 0.016 B	0.242 ± 0.016 A	0.229 ± 0.019 B
**Fv/Fm**	0.776 ± 0.005 A	0.768 ± 0.005 B	0.762 ± 0.006 C
**Fv/Fo**	3.484 ± 0.103 A	3.333 ± 0.097 B	3.228 ± 0.106 C
**Fv**	0.78 ± 0.047 B	0.800 ± 0.038 A	0.734 ± 0.047 C
**Y(NO)**	0.224 ± 0.005 C	0.232 ± 0.005 B	0.238 ± 0.006 A
**SPAD**	46.3 ± 2.359 A	42.6 ± 2.3 C	45.1 ± 2.9 B
**NO.**	86	60	112

Different letters indicate significant difference between genotypes (LSD test, *p* < 0.05).

## Data Availability

All the datasets are presented in the manuscript and Appendix A.

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
