# Peer review of "Analysis of Photosynthetic Characteristics and Screening High Light-Efficiency Germplasm in Sugarcane"

_plants, 2024, doi:10.3390/plants13050587_

Round 1

Reviewer 1 Report

Comments and Suggestions for Authors

The paper has potential, but data presentation is quite poor especially Tables 3 and 4, making it almost impossible to assess the paper. Please consider more accessible data presentation.

Author Response

Comment: The paper has potential but data presentation is quite poor especially Tables 3 and 4. making it almost impossible to assess the paper. Please consider more accessible data presentation.

Response: Thank you for the positive feedback. Your suggestion has been followed. Table 3 has been moved to a supplementary file.

Reviewer 2 Report

Comments and Suggestions for Authors

A good piece of research, demonstrating practical significance in the realm of sugarcane cultivation, whether in China or elsewhere.

Comments on the Quality of English Language

The manuscript would benefit from overall improvement in English language usage, including clarity, coherence, and precision in expression. 

Author Response

Abstract:

Provide a concise description of the experimental site location and agronomic practices,

emphasizing the global importance of sugarcane.

Further, integrate specific details about the germplasm screening process and any unique

methodology to enhance the abstract.

Add a sentence explicitly stating the gap in current knowledge or the novelty of your study.

Specify which aspect of chlorophyll fluorescence in sugarcane has not been extensively explored

before.

Response: Thank you for the suggestion. We have revised the abstract accordingly and included each point you mentioned. Please see the revised abstract in the manuscript in track changes.

Introduction:

Strengthen the written English in this section.

Response: Thank you for the suggestion. We have revised the introduction section thoroughly.

Establish a clearer connection between the global significance of sugarcane and the specific

focus on photosynthesis. Clearly articulate how improving photosynthetic efficiency aligns with

addressing challenges in the industry.

Response: Thank you for the suggestion. The revision has been made accordingly in the introduction section in lines 52-69.

Clarify research objectives and explicitly state the significance of comprehensively analyzing

chlorophyll fluorescence parameters.

Response: Thank you for the suggestion. The revision has been made accordingly in the revised manuscript in lines 112-115.

Materials and Methods:

Provide a brief description of the experimental site location, agro-climatic conditions, and

agronomic practices throughout the experiment.

Response: Thank you for the suggestion. The revision has been made accordingly in the revised manuscript in lines 332-334.

Add scientific units to the parameters taken throughout.

Response:

Thank you for your suggestions. We have made corresponding revisions to the manuscript at line 342.

Discussion:

Strengthen the argumentation with relevant citations.

Ensure consistency in terminology throughout; the language is generally formal and suitable for

a scientific audience."

Response: Thank you for the suggestion. We have updated the revised manuscript with relevant and recent literature.

Reviewer 3 Report

Comments and Suggestions for Authors

The manuscript by Wei et al recorded the chlorophyll fluorescence parameters and chlorophyll content of 258 sugarcane genotypes through 3 years of field experiments, according to which the sugarcanes were classified into 3 categories. The paper also linked high photosynthetic efficiency genotypes with high yield and high sucrose content that can guide sugarcane breeding and production.

Please describe in the Materials and Methods section or a supplementary figure how leaf positions are defined. And specific sites used to test chlorophyll fluorescence and chlorophyll content.

Author Response

Please describe in the Materials and Methods section or a supplementary figure how leaf positions are defined.And specific sites used to test chlorophyll fluorescence and chlorophyl content.

Response: Thank you for the suggestion. We have added a supplementary figure explaining the different leaf positions and sites used to test the parameters.

Please see line 351-352.